# Inflammation Confers Healing Advantage to Corneal Epithelium Following Subsequent Injury

**DOI:** 10.3390/ijms24043329

**Published:** 2023-02-07

**Authors:** Jin Suk Ryu, So Yeon Kim, Mee Kum Kim, Joo Youn Oh

**Affiliations:** 1Laboratory of Ocular Regenerative Medicine and Immunology, Biomedical Research Institute, Seoul National University Hospital, 101 Daehak-ro, Jongno-gu, Seoul 03080, Republic of Korea; 2Department of Ophthalmology, Seoul National University College of Medicine, 103 Daehak-ro, Jongno-gu, Seoul 03080, Republic of Korea

**Keywords:** cornea, epithelium, inflammatory memory, injury, memory, stem/progenitor cells, wound healing

## Abstract

Recent evidence shows that epithelial stem/progenitor cells in barrier tissues such as the skin, airways and intestines retain a memory of previous injuries, which enables tissues to accelerate barrier restoration after subsequent injuries. The corneal epithelium, the outermost layer of the cornea, is the frontline barrier for the eye and is maintained by epithelial stem/progenitor cells in the limbus. Herein, we provide evidence that inflammatory memory also exists in the cornea. In mice, eyes that had been exposed to corneal epithelial injury exhibited faster re-epithelialization of the cornea and lower levels of inflammatory cytokines following subsequent injury (either the same or a different type of injury) relative to naïve eyes without previous injury. In ocular Sjögren’s syndrome patients, corneal punctate epithelial erosions were significantly reduced after experiencing infectious injury compared with before. These results demonstrate that previous exposure of the corneal epithelium to inflammatory stimuli enhances corneal wound healing in response to a secondary assault, a phenomenon which points to the presence of nonspecific inflammatory memory in the cornea.

## 1. Introduction

Immune memory, which is the capacity to respond more rapidly and effectively to pathogens that have been encountered previously, originally was identified in cells of the adaptive immune system (T and B lymphocytes) [1,2] and later in innate immune cells (macrophages, natural killer cells and innate lymphoid cells) [2,3,4]. Recently, the memory paradigm has been expanded to encompass cells outside the immune system, most notably stem/progenitor cells (SCs) in epithelial barriers of the skin, intestines and airways [2,5,6]. In a pioneering study by Naik et al. [7], mouse skin epidermal SCs that had been exposed to inflammatory insults caused by chemical, mechanical or infectious injury were found to have harbored a memory of the event and to have acted more quickly and robustly upon a secondary assault than to the primary insults, leading to accelerated skin wound healing and hastened barrier restoration. Similar phenomena have been observed with human respiratory epithelial SCs [8], murine intestinal epithelial SCs [9] and murine hair follicular SCs [10]: retention of inflammatory memory in barrier epithelial SCs upon transient inflammatory exposure enabled the cells to protect barrier tissues from subsequent insults.

The cornea, the outermost part of the eye, is the frontline ocular barrier, and the corneal epithelium, directly exposed to the external environment, is particularly prone to various noninfectious and infectious injuries. Therefore, corneal epithelial wound healing is integral to the maintenance of corneal clarity and protection of the intraocular tissues. Central to corneal wound healing are corneal epithelial SCs located in the limbus called limbal epithelial SCs [11,12]. In this study, we investigated whether injury to the corneal epithelium confers inflammatory memory that promotes wound healing following subsequent related or unrelated injuries in both mice and human patients.

## 2. Results

### 2.1. Previous Exposure to Injury Enhances Corneal Wound Healing upon Subsequent Assault Different from Previous Injury

To address whether injury to the corneal epithelium exerts a lasting influence on the cornea following subsequent injury, we made an abrasion injury to the cornea (primary injury) in 8-week-old BALB/c mice by scraping off the central 2 mm diameter corneal epithelium, while keeping the epithelial basement membrane intact, with a surgical blade (day 0). In agreement with our previous observation [13], the corneal epithelium spontaneously healed without scarring in all mice by day 7–14, and inflammation completely resolved by day 28 (Figure 1) as evidenced by absence of neutrophils and apoptotic cells in the cornea (Figure 1B) and normalization of inflammatory cytokine levels (Figure 1C).

On day 28, we challenged the injury-recovered cornea with chemical/mechanical assaults (secondary injury), applying absolute ethanol to the whole corneal surface for 15 s and removing its epithelium. Mice that had received chemical/mechanical injury without previous abrasion injury served as positive controls, and those without either primary or secondary injury as negative controls. The eyes were daily observed under slit-lamp biomicroscopy and imaged using corneal photography. Seven days after secondary injury (day 35), the corneas were extracted for molecular assays (Figure 2A).

Corneal re-epithelialization was significantly faster following secondary injury in mice that had been subjected to primary injury when compared with the positive controls without primary injury (Figure 2B–D). Seven days after secondary chemical/mechanical injury (day 35), the area of corneal epithelial defect was only 18.0 ± 11.1% (relative to total corneal area) in mice previously exposed to primary injury, whereas it was 64.8 ± 15.8% in naïve mice without previous injury (*p* < 0.0001) (Figure 2D). Similarly, the mRNA levels of proinflammatory cytokines, IL-1β, IL-6 and MMP9, as measured using RT–qPCR, were significantly lower in the corneas that had been exposed to primary injury than in those that had not (Figure 2E).

These results suggest that exposure of the corneal epithelium to inflammatory stimuli enables it to heal better when subsequently injured through acceleration of re-epithelialization and downregulation of inflammation.

### 2.2. Previous Exposure to Injury Promotes Corneal Wound Healing Following Subsequent Encounters with Same Injury

We next determined whether the healing potential of a previously injured cornea also is enhanced after subsequently being subjected to the same assault. To this end, 28 days after inducing a corneal abrasion in mice (primary injury), we rechallenged the mice with the same type of injury (abrasion of the central 2 mm diameter corneal epithelium) (secondary injury) and assessed the eyes for wound healing and inflammatory response (Figure 3A).

Consistent with the aforementioned data (Figure 2), the corneal epithelium healed more rapidly following secondary challenge in mice that had experienced primary injury than in naïve mice not previously exposed to injury (Figure 3B,C). Corneal re-epithelialization was completed within 7 days of secondary injury in all mice with primary injury, whereas corneal epithelial healing was complete in only 20% of mice without previous injury (Figure 3C). The corneal levels of IL-1β, IL-6 and MMP9 after secondary injury also were reduced in mice that had received primary injury, as compared with naïve mice that had not (Figure 3D).

Together, the data demonstrate that injury to the corneal epithelium allows the cornea to heal the wound and restore the ocular barrier more quickly upon subsequent injuries, regardless of whether the subsequent injury is the same as or different from the previous injury.

### 2.3. Ocular Sjögren’s Syndrome (SjS) Patients Exhibit Reduced Punctate Epithelial Erosions (PEEs) after Corneal Infection

To evaluate the beneficial effects of transient exposure to inflammatory stimulus on the healing capacity of the corneal epithelium in a clinical setting, we examined a cohort of patients with ocular SjS. Out of 929 patients who had been followed-up for primary or secondary ocular SjS at Seoul National University Hospital (Seoul, Republic of Korea) from 2010 to 2020, we identified 11 who had experienced infectious keratitis transiently over the disease course and had been consecutively followed up for ocular SjS-related corneal PEEs before and after infection. The demographical, general medical and ocular characteristics of the patients are shown in Table 1. 

All were adult women aged 68 ± 11 years at the time of corneal infection. The infection occurred 1612.5 ± 1487.8 days after ocular SjS diagnosis and was resolved in all patients within the mean 37.5 days of medical treatment. We sequentially reviewed the relevant medical records and anterior segment photographs with fluorescein dye staining for each individual patient and evaluated the corneal PEEs before infection (the last follow-up before infection occurred) and after infection (one week after infection was resolved). Corneal PEE grading was carried out based both on ocular staining scores using Sjögren’s International Collaborative Clinical Alliance (SICCA) Ocular Staining Score (OSS) and corneal staining scores on the National Eye Institute (NEI)/Industry Workshop Scale [14].

Comparative analysis of corneal PEEs before and after infection in the same individual patient revealed that the severity of corneal epitheliopathy (as indicated by the OSS and NEI scales) was significantly decreased after the cornea experienced infection, as compared with before infection developed, in all of the patients (Figure 4A,B, Table 1). It should be noted that there were no differences in topical medications affecting corneal PEEs between before and after infection except for topical corticosteroids: 5 out of 11 patients had used topical corticosteroids for ocular SjS-related inflammation control before infection, whereas no patient used it after infection. (Table 1). The ocular staining scores by SICCA OSS were 8.2 ± 3.1 (2–12) before infection developed and 6.0 ± 3.0 (1–9) after infection was resolved (*p* = 0.001, Wilcoxon matched-pairs signed rank test). Correspondingly, the corneal staining scores on the NEI scale also were reduced from 9.9 ± 3.5 (2–13) before infection to 7.2 ± 3.6 (1–11) after infection (*p* = 0.001, Wilcoxon matched-pairs signed rank test). Moreover, corneal filaments were present in 9 out of 11 (82%) patients before infection, whereas they were found in 6 (55%) patients after infection episodes (Table 1). Given the fact that the corneal epithelium in ocular SjS patients is constantly subjected to SjS-induced desiccating injury manifesting as PEEs, our observation that corneal PEEs in ocular SjS patients were abrogated after exposure to infectious injury indicates that the inflammatory stimuli to the cornea caused by infection augmented the healing potential of the corneal epithelium following subsequent desiccating injury.

## 3. Discussion

In this study, we showed that a primary inflammatory stimulus to the corneal epithelium conferred a corneal capacity to accelerate healing and ocular barrier restoration upon subsequent insults. This memory response was nonspecific: that is, it was effective both when a secondary challenge was the same as the previous injury and when it was different.

Our findings add to the emerging evidence of inflammatory memory in barrier tissues [2,5,6]. In a pioneering work by Naik et al. [7], an acute exposure of the skin to a psoriatic-like inflammatory stimulus endowed murine epidermal SCs with epigenetic memories of their inflammatory experiences, thus enabling the epidermis to heighten the wound healing response to various types of subsequent injuries. Moreover, in their study, infectious injury (with the fungal pathogen *Candida albicans*) as well as noninfectious injuries also triggered accelerated wound healing response in the skin following a secondary wounding injury [7]. This is in line with our observation that the episode of corneal infection in ocular SjS patients led to the attenuation of corneal epitheliopathy in response to subsequent desiccating stress. In addition to epidermal SCs, murine hair follicle SCs, upon an epidermal denuding injury, were shown to migrate to repair the damaged epidermis and accumulate long-lasting epigenetic memories, thereby enhancing the regenerative capacity of the epidermis to future assaults [10]. Human respiratory epithelial SCs isolated from chronic rhinosinusitis were also found to retain allergic inflammatory memory at the chromatin level through long-term expansion in a culture [8]. In murine intestines, transient in utero exposure to circulating maternal IL-6 and the ensued inflammation caused epigenetic rewiring of fetal intestinal epithelial cells that persisted to adulthood, thereby providing enhanced protection against Salmonella oral infection [9]. In addition to barrier epithelial SCs, stromal cells including fibroblasts [15], pancreatic epithelial cells [16,17] and Schwann cells [18] have demonstrated inflammatory memory.

Mechanistically, there are several possibilities for enhanced corneal wound repair after inflammation. One entails the epigenetic memories encoded at the level of chromatin in corneal limbal epithelial SCs, as previous studies on epithelial SCs have elucidated. Following an initial inflammatory stimulus, cells modify their histones and chromatins in order to make stress-related genes accessible to transcription factors and become activated, while some of the chromatins remain accessible, thereby permitting rapid transcriptional activation upon a secondary challenge [2,5,6]. It is also possible that other cells in the cornea, in addition to epithelial SCs, might be involved in memory. Corneal wound repair is a complex process coordinated by multiple cell types such as corneal epithelial cells, corneal stromal fibroblasts, resident and infiltrating immune cells, and corneal nerves [19,20]. As aforementioned, diverse cell types recently have been found capable of retaining inflammatory memory upon injury. Therefore, it seems more plausible that, rather than acting individually, these multiple corneal cells engage in concert to encode memory and promote adaptation to environmental stress in the cornea. Further studies into the underlying mechanism of inflammatory memory at cellular levels in the cornea are required.

It is important to note that inflammatory memory can be maladaptive and have detrimental implications, depending on context. If a memory of acute inflammation lingers on, aberrant or excessive inflammation in response to mild triggers can ensue, leading to chronic and recurrent corneal inflammation, scarring or even dysplasia. Moreover, hyperactive and precocious activation of limbal epithelial SCs may result in SC exhaustion, thereby accelerating aging. Indeed, respiratory epithelial SCs have been reported to form memories of chronic exposure to an allergic inflammatory environment and propagate disease, which established the paradigm of SC dysfunction [8]. In the intestinal epithelium, a sustained high-fat diet was found to increase the susceptibility of intestinal epithelial SCs to spontaneous intestinal dysplasia and carcinomas [21].

Taken together, our results implicate the presence of nonspecific inflammatory memory in the cornea that, in the present case, conferred a healing advantage to the corneal epithelium following subsequent injury. Future research will unfold the underlying mechanisms of corneal inflammatory memory and provide new therapeutic strategies by which its utility in promoting tissue adaptation and controlling maladaptive memories can be maximized.

## 4. Materials and Methods

### 4.1. Animals and Animal Models

The experimental protocol was approved by the Institutional Animal Care and Use Committee of the Seoul National University Biomedical Research Institute (Seoul, Republic of Korea), and the animal experiments were performed according to the ARVO Statement for Use of Animals in Ophthalmic Vision and Research.

Seven-week-old male BALB/c mice were purchased from KOATECH (Pyeongtaek, Republic of Korea) and housed in a specific pathogen-free environment. After a one-week adaptation period, the mice were subjected to primary injury in the form of corneal abrasion. Specifically, the central cornea was marked with a 2 mm trephine, and its epithelium was scraped off with a #10 surgical blade under an operating microscope. Twenty-eight days later, the mice were subjected to secondary injury: either chemical burn followed by mechanical epithelial removal (a different injury from the primary injury) or a central 2 mm diameter epithelial abrasion (the same as the primary injury). The chemical burn was induced by applying absolute ethanol to the whole corneal surface for 15 s, and after thoroughly rinsing with 2 mL phosphate-buffered saline, the corneal epithelium was removed with a #10 surgical blade. Mice that had received primary injury, but not secondary injury, served as positive controls, and those that had received neither were used as negative controls.

### 4.2. Clinical Examination for Corneal Epithelial Defect

The mice were observed daily for corneal epithelial defect under slit-lamp biomicroscopy and photographed three times per week with a camera mounted on a surgical operating microscope. For quantitation of corneal epithelial defect, corneal photographs were taken at the same magnification after lissamine green vital dye staining (Sigma-Aldrich, Saint Louis, MO, USA), and the proportion of the stained area relative to the total corneal surface (%) was calculated using ImageJ software (NIH, Bethesda, MD, USA).

### 4.3. Histology

For histologic assays, mice were humanely killed, and the corneas were excised. The excised cornea was fixed in 10% (v/v) formaldehyde, paraffin-embedded, and cut into 4-μm sections. The sections were subjected to hematoxylin–eosin staining, terminal deoxynucleotidyl transferase-mediated nick end labeling (TUNEL) staining (ApopTag^®^ Red in situ Apoptosis Detection Kit, EMD Millipore, Billerica, MA, USA) or neutrophil immunostaining. For neutrophil staining, a rat anti-mouse neutrophil antibody (ab2557; Abcam, Cambridge, MA, USA) and a goat anti-rat IgG TRITC (AP136R, Sigma-Aldrich) were used as primary antibody and secondary antibody, respectively. DAPI solution was used for counterstaining (IHC WORLD, Woodstock, MD, USA).

### 4.4. Real-Time Reverse Transcription Quantitative Polymerase Chain Reaction (RT–qPCR)

The excised corneas were cut into small pieces, incubated in RNA isolation reagent (RNA-Bee, Tel-Test, Inc., Friendswood, TX, USA), and sonicated with an Ultrasonic Processor (Cole Parmer Instruments, Vernon Hills, IL, USA). RNA was extracted using the RNeasy Mini kit (Qiagen, Valencia, CA, USA) and converted to first-strand cDNA by reverse transcription (High Capacity RNA-to-cDNA^TM^ Kit, Applied Biosystems, Carlsbad, CA, USA). Real-time amplification was performed in TaqMan^®^ Universal PCR Master Mix (Applied Biosystems) in an automated instrument (ABI 7500 Real Time PCR System, Applied Biosystems). The PCR probe sets were purchased from Applied Biosystems (TaqMan^®^ Gene Expression Assay kits, Applied Biosystems). Data were normalized to mouse GAPDH, analyzed using the 2^−∆∆Ct^ method, and recorded as fold changes of mRNA levels relative to the negative controls.

### 4.5. Patients and Evaluations

This study was approved by the Institutional Review Board (IRB) of Seoul National University Hospital (Seoul, Republic of Korea) (IRB No. 2012-098-1181) and conducted with adherence to the Declaration of Helsinki. Due to the retrospective nature of this study and under IRB approval, patient consent was waivered.

A total of 929 patients who had been consecutively followed-up at Seoul National University Hospital (Seoul, Republic of Korea) between 2010 and 2020 after diagnosis of primary or secondary ocular SjS were considered. Among them, 11 patients who had experienced a transient episode of infectious keratitis during the follow-up period were identified and further evaluated. Anterior segment photographs (with and without fluorescein vital dye staining) as well as medical charts were consecutively evaluated for each patient. Ocular surface staining scoring was performed on the basis of both medical records and fluorescein-stained anterior segment photographs. The SICCA OSS and NEI/Industry Workshop Scale were used to grade the corneal/conjunctival staining (on a scale from 0 to 12) and corneal staining (on a scale of 0 to 15), respectively [14]. The staining scores were compared for the same individual between the last follow-up before infection occurred and one week after infection was resolved (as defined by disappearance of corneal epithelial defect and stromal infiltration).

### 4.6. Statistical Analysis

Prism software v.9.4.0 (GraphPad, San Diego, CA, USA) was used for statistical testing and graph generation. The Shapiro–Wilk test or Kolmogorov–Smirnov test was used to determine a normal distribution of data in each group. One-way ANOVA with Tukey’s test was applied for comparison of mean values from more than two groups. The Student *t*-test, Mann–Whitney U test and Fisher’s exact test were used for comparison of the two groups. The Wilcoxon matched-pairs signed rank test was used to compare the means of the pre- and post-infection values in the same patient. Data were presented as mean ± SD. Differences were considered significant at *p* < 0.05.

## Figures and Tables

**Figure 1 ijms-24-03329-f001:**
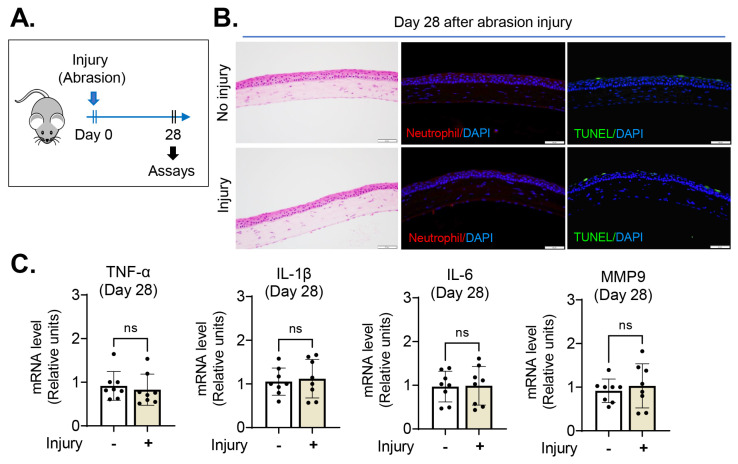
The cornea completely recovered from epithelial abrasion injury by day 28. (**A**) Experimental scheme: the central 2 mm diameter corneal epithelium was scraped off (day 0), and the cornea was assessed for re-epithelialization and inflammation on day 28. (**B**) Representative microphotographs of the cornea with hematoxylin–eosin staining, neutrophil immunostaining and TUNEL staining; scale bar: 50 μm. (**C**) RT–qPCR for proinflammatory cytokine levels in the cornea: The *y*-axis denotes fold changes in mRNA levels relative to controls without injury. Mean values ± SD are shown, and a dot depicts the data from an individual mouse (eight animals per group). ns: not significant as analyzed using Mann–Whitney U test (TNF-α) or Student’s *t*-test (IL-1β, IL-6, MMP9).

**Figure 2 ijms-24-03329-f002:**
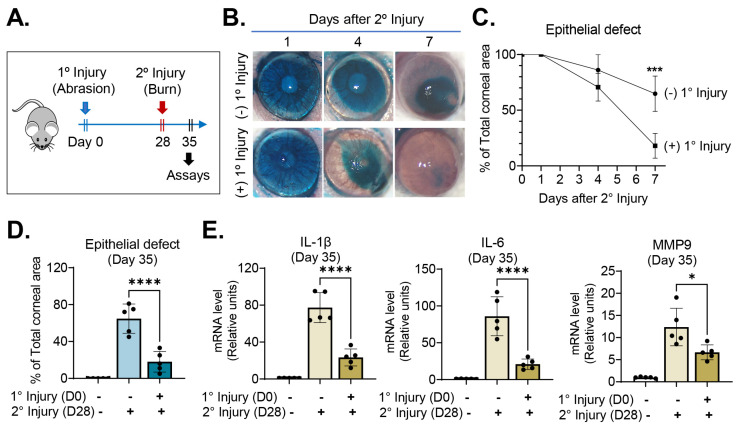
Previous abrasion injury enabled the cornea to heal wounds faster following subsequent burn injury. (**A**) Experimental schematic: twenty-eight days after a self-resolving abrasion injury (day 28), the cornea was subjected to secondary challenge in the form of a burn injury. (**B**) Representative corneal photographs after lissamine green vital dye staining: The green-stained areas are portions of the corneal epithelial defects. Following secondary injury, corneal re-epithelialization was faster in eyes with primary injury than in those without. (**C**) Time course of corneal re-epithelization after secondary burn injury: the *y*-axis denotes the % of corneal epithelial defect area relative to the total corneal area. (**D**) Quantitation of corneal epithelial defect 7 days after secondary burn injury (day 35). (**E**) RT–qPCR for proinflammatory cytokine levels in the cornea 7 days after secondary injury (day 35): the *y*-axis denotes fold changes in mRNA levels relative to negative controls not subjected to any injury. Mean values ± SD are shown, and a dot depicts the data from an individual mouse (five animals per group). * *p* < 0.05, *** *p* < 0.001, **** *p* < 0.0001, as analyzed using Student’s *t*-test (**C**) or one-way ANOVA with Tukey’s multiple comparison test (**D**,**E**).

**Figure 3 ijms-24-03329-f003:**
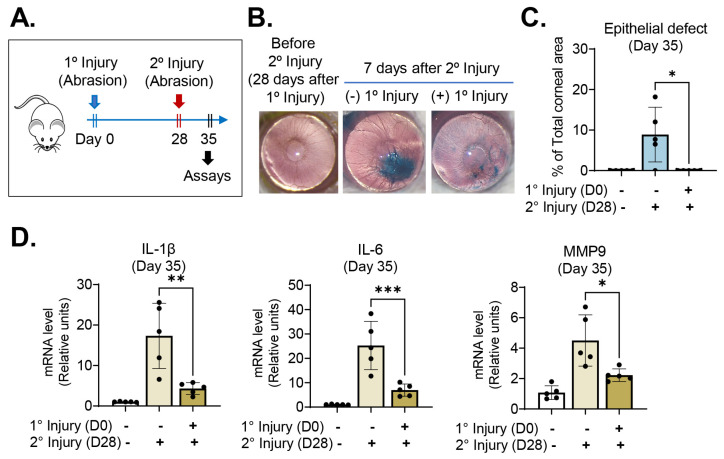
Previous abrasion injury accelerated corneal wound healing upon secondary challenge with same injury. (**A**) Experimental scheme: the cornea was rechallenged with abrasion injury 28 days after receiving primary abrasion injury (day 28). (**B**) Representative corneal photographs after lissamine green staining 7 days after secondary challenge (day 35): The green-stained areas correspond to corneal epithelial defects. Re-epithelialization was complete in the cornea previously exposed to injury, whereas epithelial defect remained in the cornea without previous injury. (**C**) Measurement of corneal epithelial defect area (% of total corneal area) on day 7 after secondary injury (day 35). (**D**) mRNA levels of proinflammatory cytokines in cornea 7 days after secondary injury (day 35): The *y*-axis represents fold changes of mRNA levels relative to normal corneas without injury. Mean values ± SD are shown. A single dot depicts the data from an individual animal (5 animals per group). * *p* < 0.05, ** *p* < 0.01, *** *p* < 0.001, as analyzed using Mann–Whitney U test (**C**) or one-way ANOVA with Tukey’s multiple comparison test (**D**).

**Figure 4 ijms-24-03329-f004:**
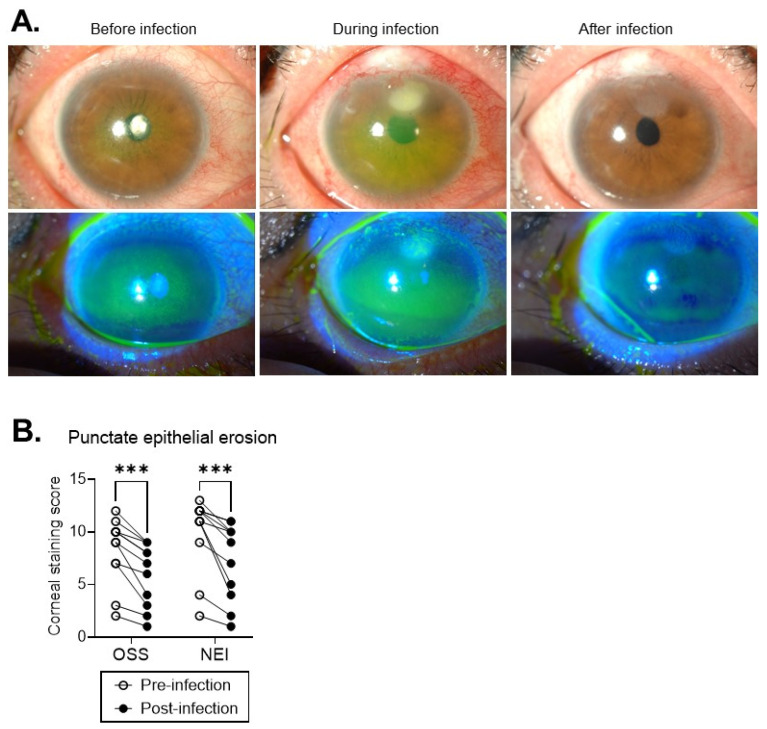
Exposure to infection episode attenuated corneal epitheliopathy in ocular SjS patients post-infection relative to pre-infection. (**A**) Representative anterior segment photographs of ocular SjS patient before, during and after infectious keratitis: the green-stained punctate areas in the fluorescein-stained images (lower row) reflect PEEs in the cornea. (**B**) The amounts of corneal punctate epithelial erosions as graded by SICCA Ocular Staining Score (OSS) and the National Eye Institute (NEI)/Industry Scale in each patient before infection occurred and after infection subsided. *** *p* = 0.001, as analyzed by Wilcoxon matched-pairs signed rank test (Pre- vs. Post-infection).

**Table 1 ijms-24-03329-t001:** Clinical and ocular characteristics of ocular SjS patients before vs. after corneal infection.

Demographical and General Medical Characteristics			
No. of patients (eyes)		11 (11)	
Age (years)		68 ± 11 (50–86)	
Sex (female:male)		11:0	
Time after SjS diagnosis (days)		1612.5 ± 1487.8	
Salivary gland involvement		3 (27%)	
Primary SjS:secondary SjS		7:4	
Diabetes mellitus		3 (27%)	
Systemic immunosuppressant use		8 (73%)	
Ocular Management (Pre- vs. Post-infection)			
	Before infection	After infection	*p* value
Autologous serum or artificial tears	11	11	>0.999 *
Topical corticosteroids	5	0	0.035 *
Topical antiglaucoma medication	3	2	>0.999 *
Ocular Conditions (Pre- vs. Post-infection)			
	Before infection	After infection	*p* value
Corneal filaments	9 (82%)	6 (55%)	0.361 *
Punctate epithelial erosions SICCA Ocular Staining Score NEI/Industry Workshop Scale	8.2 ± 3.1 9.9 ± 3.5	6.0 ± 3.0 7.2 ± 3.6	0.001 ** 0.001 **
Corneal Infection Characteristics			
Bacteria:fungus:unidentified		4:1:2	
Treatment period (days after infection diagnosis to resolution)		37.5 ± 83.8	

* Fisher’s exact test; ** Wilcoxon matched-pairs signed rank test.

## Data Availability

The data presented in this study are available within the article.

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
