# Peer review of "Inflammation Confers Healing Advantage to Corneal Epithelium Following Subsequent Injury"

_ijms, 2023, doi:10.3390/ijms24043329_

Round 1
Reviewer 1 Report
The study documents inflammatory memory present in the cornea. The issue is interesting, especially that there are no many publications int this area.
Regarding the human study, 11 patient with SjS were analysed. It was stated that the cornea had attenuated epitheliopathy after the infection. However, it is not clear when the cornea was evaluated (time after infection). Is it possible that the intensive treatment received for infection attenuated the inflammation and attenuated the epithelium disease for some time?
REgarding the animal study, the number of animals investigated should be provided.
Author Response
1. Regarding the human study, 11 patient with SjS were analysed. It was stated that the cornea had attenuated epitheliopathy after the infection. However, it is not clear when the cornea was evaluated (time after infection).
→ Reply: We evaluated corneal PEEs (post-infection) one week after infection was completely resolved (as defined by complete resolution of epithelial defect and stromal infiltration). Per reviewer's comment, we added this detail in the revised manuscript (Line 164-165, 328-330).
2. Is it possible that the intensive treatment received for infection attenuated the inflammation and attenuated the epithelium disease for some time?
→ Reply: To address the reviewer's concern, we looked into topical medications that patients received and compared them before and after infection. There were no differences in the use of artificial tears, autologous serum eye drops and anti-glaucoma drops between pre- and post-infection (Table 1 of the revised manuscript, Line 172-176), except for the use of topical steroids. Five out of 11 patients had used topical corticosteroids before infection for ocular SjS-related inflammation control, while no patient used it after infection. Considering that topical steroids are used for control of inflammation and treatment of corneal epitheliopathy in ocular SjS patients, it can be ruled out that the medication might affect the attenuation of corneal epitheliopathy in patients after infection. Also, most of our patients used fortified antibiotic eye drops for infection control, and it is well-known that fortified antibiotics (formulated in a hospital pharmacy) cause corneal epitheliopathy. Thus, the intensive treatment (fortified antibiotic eye drops) during infection might have had a detrimental effect on corneal epitheliopathy, let alone a beneficial effect. We thank the reviewer for bringing up this impoertant issue and giving us the opportunity to address it.
3. Regarding the animal study, the number of animals investigated should be provided.
→ Reply: We have added the number of animals used for each experiment in the legend of each figure of the revised manuscript (Line 73, 103-104, 142-143).
Reviewer 2 Report
1. The title of this manuscript emphasis on inflammatory state, however, the Abstract and Discussion parts emphasis inflammatory memory. It seems to be contradictory.
2. It is necessary to clarify inflammatory state or inflammatory memory, and the inflammation at 28 days after 1° injury is an important reference indicator. However, in Figure 1 and 2, the author only detected the inflammatory cytokines on the 35th day. So, we suggest that the inflammatory cytokines on the 28th day must be detected.
3.Fig.1B reflects the healing course of two groups, but the slit lamp images of (-) 1° injury at day 4 and day 7 showed a significant difference. Part of the cornea has re-epithelialized on the fourth day, but was severely damaged again on the seventh day. It is necessary to make sure the slit lamp images are from the same mouse.
4. According to the authors, inflammatory memory promotes the healing of corneal epithelium, however, it is not clear whether the corneal epithelial phenotype changes after healing or not. It is better to further investigate the epithelial phenotype.
5. In the animal experiments, the author intended to make it clear that inflammation/injury memory promotes the healing of corneal epithelium. However, it seems that inflammatory/impaired memory did not involve in the selected clinical cases. In addition, it could not be ruled out whether the improvement of punctate epithelial erosions was related to drug therapy after corneal infection. SjS patient is not a suitable choice for this research.
6. According to the Introduction and Discussion, the "memory" introduced in the manuscript is mainly derived from immune cells or epithelial stem/progenitor cells (SCs). However, only three inflammatory cytokines have been detected and no cell-related research were conducted.
Author Response
1. The title of this manuscript emphasis on inflammatory state, however, the Abstract and Discussion parts emphasis inflammatory memory. It seems to be contradictory.
→ Reply: The authors humbly submit that the title in its current form ("Inflammation confers healing advantage to corneal epithelium following subsequent injury") implies a concept of inflammatory memory because it literally indicates that primary inflammation endows the cornea with healing power upon a secondary assault. Please understand that we determined the title of this paper so that it could directly describe its findings.
2. It is necessary to clarify inflammatory state or inflammatory memory, and the inflammation at 28 days after 1°injury is an important reference indicator. However, in Figure 1 and 2, the author only detected the inflammatory cytokines on the 35th day. So, we suggest that the inflammatory cytokines on the 28th day must be detected.
→ Reply: Totally agreeing with the reviewer, we confirmed that inflammation was completely resolved and the cornea completely recovered from the first abrasion injury on day 28 through clinical, histological and molecular examinations, and added these data in the revised manuscript (Fig 1, Line 62-74, 289-298). On day 28 (before the secondary injury was made), the corneal epithelial and stromal structures were normal (Fig 1B), there were no inflammatory cells or apoptotic cells in the cornea (Fig 1B), and the levels of key inflammatory cytokines in the cornea returned to normal, baseline levels (Fig 1C). We thank the reviewer for bringing up this important issue and giving us the opportunity to address it.
3. Fig.1B reflects the healing course of two groups, but the slit lamp images of (-) 1° injury at day 4 and day 7 showed a significant difference. Part of the cornea has re-epithelialized on the fourth day, but was severely damaged again on the seventh day. It is necessary to make sure the slit lamp images are from the same mouse.
→ Reply: The images of day 1, 4, 7 in each group (Fig. 1B) were serially taken from the same animal. The confusion was created because the orientation of the eye in photographs was not arranged in the same direction from picture to picture. We corrected this error in the revised figure (Fig. 2B of the revised manuscript).
4. According to the authors, inflammatory memory promotes the healing of corneal epithelium, however, it is not clear whether the corneal epithelial phenotype changes after healing or not. It is better to further investigate the epithelial phenotype.
→ Reply: In the present study, we confirmed that the corneal epithelial healing was accelerated in the cornea previously exposed to the inflammatory injury mainly based on clinical observation (Fig. 2B-D, Fig 3B, C). Per the reviewer's comment, we admit the need for further investigation at cellular levels, and added this aspect to the discussion (Line 242-243). Our lab is planning to do these experiments and hope to get the publishable data in the next few years. We thank the reviewer for the valuable suggestion.
5. In the animal experiments, the author intended to make it clear that inflammation/injury memory promotes the healing of corneal epithelium. However, it seems that inflammatory/impaired memory did not involve in the selected clinical cases. In addition, it could not be ruled out whether the improvement of punctate epithelial erosions was related to drug therapy after corneal infection. SjS patient is not a suitable choice for this research.
→ Reply: To address the reviewer's concern, we looked into topical medications patients received before and after infection and compared them. There were no differences in the use of artificial tears, autologous serum eye drops and anti-glaucoma medications between pre- and post-infection (Table 1 of the revised manuscript, Line 172-176). One exception was topical steroids. Five out of 11 patients had used topical corticosteroids before infection for ocular SjS-related inflammation control, while no patient used it after infection. Given that topical steroids attenuate corneal epitheliopathy through suppressing inflammation in ocular SjS patients, the possibility can be ruled out that the medication affected the attenuation of corneal epitheliopathy in patients after infection. Moreover, most of our patients were treated with fortified antibiotic eye drops for infection control, and it is well-known that fortified antibiotics (formulated in a hospital pharmacy) rather cause corneal epitheliopathy. Thus, the intensive treatment (fortified antibiotic eye drops) during infection might have had a detrimental effect on corneal epitheliopathy, let alone a beneficial effect.
While admitting that SjS patients might not be a perfect choice for this research, we reckon that our clinical data (Sjogren's syndrome patients with a history of infection) have implications in two ways. First, it suggests the possibility that infectious injury (as well as non-infectious injury) can also induce accelerated wound healing in the cornea following a secondary injury. This is consistent with the findings by Naik et al. (Nature 2017, 550, 475-480): in their animal study, infectious injury (with the fungal pathogen Candida albicans) as well as non-infectious injuries also triggered an augmented wound healing response in the skin following a secondary wounding injury. We added this discussion to the revised manuscript (Line 38, 216-128). Second, our finding has clinical implication as ocular SjS patients have higher incidence of infectious keratitis and are constantly exposed to desiccating stress injury. In fact, our experience in clinic suggests that ocular discomfort as well as corneal PEEs was often reduced in ocular SjS patients after they had experienced corneal infection. The inflammatory memory (or accelerated healing response of the corneal epitheium after inflammation as we presented in our paper) can be one of the reasons underlying this clinical phenomenon. Hence, we humbly submit that our paper might introduce the new concept and field of research to both scientists and clinicians.
6. According to the Introduction and Discussion, the "memory" introduced in the manuscript is mainly derived from immune cells or epithelial stem/progenitor cells (SCs). However, only three inflammatory cytokines have been detected and no cell-related research were conducted.
→ Reply: As aforementioned, we agree that further studies are required to understand the underlying mechanism of inflammatory memory at cellular levels in the cornea (Line 242-243). The research on the inflammatory memory in non-immune cells is budding as we elaborated in the introduction and discussion (you can also see in our reference lists, most of which have been published in the last few years). Our lab is planning to do these experiments and hope to get the publishable data in the next few years. Also, we hope that our present paper (if published) will usher corneal researchers into this exciting, new area of research. The cornea is the frontline of the eye, and therefore well-positioned for this memory response.
Reviewer 3 Report
This paper evaluates the idea of 'stem cell memory' that may contribute to accelerated barrier recovery following corneal wounding. The authors show that corneal epithelial barrier appears to recover faster in mice that had a prior injury compared to naive mice. They also show evidence in a small clinical population of patients with Sjogren's syndrome that corneal epithelial barrier function appeared to be higher in patients after an infection. This is a nice, well-written paper with consistent findings based on findings in an animal model and a human clinical population.
-Can the authors comment in the text on whether the corneal epithelial basement membrane was damaged in their debridement model?
-Did any of the mice develop scarring after the first injury in any of the studies? The presence of a scar would likely affect later recovery.
-Please add further mention in the text regarding the lower inflammatory marker expression following ocular in mice with a previous injury and if this is likely due to the increased epithelial closure. One might expect that inflammation would be higher in animals with the earlier injury given that the tissue may be primed for combatting infection and this may take time to resolve (past the 28 days).
Author Response
1. Can the authors comment in the text on whether the corneal epithelial basement membrane was damaged in their debridement model?
→ Reply: In this scraping model, we take care not to injure the basement membrane. Per the reviewer's comment, we added the relevant text to the revised manuscript (Line 61).
2. Did any of the mice develop scarring after the first injury in any of the studies? The presence of a scar would likely affect later recovery.
→ Reply: Having the same concern as the reviewer's, we did not include the animal with scarring in this study (in this epithelial abrasion model, scarring is very rare after healing, which is the reason we chose this model as a primary injury model). Therefore, it is true that any mice did not have scarring after the first injury in this study. We added this text to the revised manuscript (Line 63).
3. Please add further mention in the text regarding the lower inflammatory marker expression following ocular in mice with a previous injury and if this is likely due to the increased epithelial closure. One might expect that inflammation would be higher in animals with the earlier injury given that the tissue may be primed for combatting infection and this may take time to resolve (past the 28 days).
→ Reply: We confirmed that inflammation was completely resolved and the cornea completely recovered from the first abrasion injury on day 28 through clinical, histological and molecular examinations, and added these data in Figure 1 of the revised manuscript (Line 62-74, 289-298). Briefly, the corneal epithelial and stromal structures were normal (Fig 1B), and there were no inflammatory cells or apoptotic cells in the cornea (Fig 1B). The levels of key inflammatory cytokines in the cornea returned towards normal by day 28 (Fig 1C).
Reviewer 4 Report
This is an interesting article about the healing process after corneal epithelium injuries and repeated cornea inflammatory response. The findings of this study could be of clinical significances to improve therapy and treatment of corneal surface.
Author Response
This is an interesting article about the healing process after corneal epithelium injuries and repeated cornea inflammatory response. The findings of this study could be of clinical significances to improve therapy and treatment of corneal surface.
→ Reply: We appreciate the interest and time that the reviewer has taken in our manuscript and the encouraging comments.
Round 2
Reviewer 2 Report
Thank you for your response to these questions.
After reading the revised manuscript carefully, although the animal experiment part has been revised more rigorously, I think the whole experimental research is still too monotonous.
In my opinion, patients with Sjogren's syndrome may have chronic inflammation on the ocular surface for a long time. This chronic inflammation is the real "primary inflammation", and infection is the real "secondary assault". Increased tearing after corneal infection and the use of anti-infective eye drops may improved ocular dryness and promoted the repair of corneal epithelium. So, the case of Sjogren's syndrome is not appropriate.
To sum up, I don't think this manuscript is fit for publication.